# Exploring the Role of Artificial Intelligence in Internet of Things Systems: A Systematic Mapping Study

**DOI:** 10.3390/s24206511

**Published:** 2024-10-10

**Authors:** Umair Khadam, Paul Davidsson, Romina Spalazzese

**Affiliations:** 1Department of Computer Science and Media Technology, Malmö University, 20506 Malmö, Sweden; umair.khadam@mau.se (U.K.); romina.spalazzese@mau.se (R.S.); 2Internet of Things and People Research Center, Malmö University, 20506 Malmö, Sweden

**Keywords:** artificial intelligence, AI, internet of things, IoT, systematic mapping, machine learning, ML

## Abstract

The use of Artificial Intelligence (AI) in Internet of Things (IoT) systems has gained significant attention due to its potential to improve efficiency, functionality and decision-making. To further advance research and practical implementation, it is crucial to better understand the specific roles of AI in IoT systems and identify the key application domains. In this article we aim to identify the different roles of AI in IoT systems and the application domains where AI is used most significantly. We have conducted a systematic mapping study using multiple databases, i.e., Scopus, ACM Digital Library, IEEE Xplore and Wiley Online. Eighty-one relevant survey articles were selected after applying the selection criteria and then analyzed to extract the key information. As a result, six general tasks of AI in IoT systems were identified: pattern recognition, decision support, decision-making and acting, prediction, data management and human interaction. Moreover, 15 subtasks were identified, as well as 13 application domains, where healthcare was the most frequent. We conclude that there are several important tasks that AI can perform in IoT systems, improving efficiency, security and functionality across many important application domains.

## 1. Introduction

The integration of AI into IoT systems is generally referred to as the Artificial Intelligence of Things (AIoT), representing the convergence of two rapidly evolving technologies [1]. AIoT defines how devices interact with each other in their environment, leading to significant advancements across various sectors such as smart cities, healthcare, agriculture and industrial automation. IoT devices such as sensors, actuators and appliances collect, exchange and process the data to meet user goals effectively. AI is employed to process and analyze the large volume of data generated by IoT devices to enhance the decision-making processes and overall efficiency of the IoT systems. The integration of these two technologies opens new opportunities for automation and intelligence across diverse application domains. Recently, the adoption of AIoT systems has accelerated due to the growth in demand for automation, real-time decision-making and predictive analytics. AIoT systems are receiving extensive attention for their potential to revolutionize industries by providing advanced analytics, autonomous operations and predictive maintenance. The interaction between AI components, such as machine learning algorithms, and IoT devices allows for real-time data collection, processing, analysis and decision-making. This integration not only enhances the functionality of IoT systems but also expands the application domains, spanning from smart homes and cities to healthcare and industrial automation.

In recent years, the integration of AI and IoT has emerged as a transformation approach to enhance system intelligence. One study [2] highlighted how AI plays a crucial role in IoT systems by processing large numbers of data and making smart, autonomous decisions. Similarly, ref. [3] also provides an overview of AI in IoT, highlighting how AI enhances the functionality of IoT systems by enabling intelligent decision-making, real-time data processing and autonomous operations. Although there is consensus about the potential of using AI in IoT systems, understanding the specific roles of AI in IoT systems and where it is most effectively utilized is essential for advancing both research and practical implementations of IoT. Many previous studies have investigated various aspects of the use of AI in IoT, but a comprehensive analysis that systematically identifies and analyzes the roles of AI within IoT systems is still lacking. In this study, we conduct a systematic mapping study [4] to uncover the various roles of AI in IoT systems. Moreover, we identify the specific application domains where AI has made impact. Thus, we aim to identify and categorize the critical tasks of AI in IoT systems, such as data preprocessing, pattern recognition, decision support and predictive analytics. The main contributions of this study are the following:Identifying the roles of AI in IoT systems and a systematic mapping of the state of the art to these roles, offering valuable insights for both researchers and practitioners.Highlighting the application domains where the potential of using AI in IoT systems has been investigated.

The study is organized as follows: Section 2 explains the methodology used for the systematic mapping study. Section 3 presents and analyzes the results. Section 4 offers a discussion of the findings, and Section 5 provides conclusions and suggests future research directions.

## 2. Method

The purpose of a systematic mapping study is to give an overview of a research area and count the contributions based on certain categories, focusing on answering specific research questions, identifying gaps, highlighting research trends, gaps and patterns in the existing literature and providing a comprehensive landscape of a broader area [4]. In other words, the purpose of a systematic mapping study is to categorize and classify existing research in a particular field. It should provide a broad overview of the research landscape, identifying areas that are well studied and those that require further investigation.

This systematic mapping study aimed to identify and analyze the roles and application domains of AI in IoT systems. However, as there are several thousand scientific publications that describe the use of AI in IoT systems, making a study of the primary research was infeasible. Consequently, we studied review articles concerning different aspects of the use of AI in IoT systems.

The methodology included a detailed search strategy across multiple databases, selection criteria for the inclusion and exclusion of articles and systematic data extraction and analysis.

### 2.1. Review Protocol

The review protocol included a search strategy, selection criteria, research questions and selection procedure. The search strategy for this systematic mapping is defined as follows:Databases: Scopus, IEEE Xplore, ACM Digital Library and Wiley Online Library;Publication Years: 2019 to 2024;Search String: (“AI” OR “Artificial intelligence”) AND (“IoT” OR “Internet of Things”) AND (“Review” OR “Survey”);Search Fields: article title, abstract, keywords.

The main reason for selecting these databases is that they are regarded as some of the most important databases in the field of computer science research. As shown in Figure 1, 464 records were found from the different databases. We found 363 records from Scopus, 88 from IEEE Xplore, 8 from Wiley and 5 from ACM Digital Library.

### 2.2. Selection Criteria

The selection criteria to review the protocol are defined as follows:

(1)Inclusion Criteria

(a)Published between January 2019 and March 2024.(b)Peer-reviewed journal article.(c)Literature review or survey.(d)Focus on the use of AI in IoT systems.

(2)Exclusion Criteria

(a)Not written in English language.(b)Articles whose full text is not available.(c)Conference papers, magazine articles, book chapters and newsletters.

Figure 1 presents the Preferred Reporting Items for Systematic Reviews and Meta-Analyses (PRISMA) flow diagram, which illustrates the identification, screening and selection of the records from the various databases for inclusion in the study. The process ensured that duplicates were removed and only relevant records, based on predefined selection criteria, were considered for detailed review. A total of 464 records were retrieved from the 4 databases, of which 78 were duplicates. After removing the duplicates, 386 records remained. Applying the exclusion criteria first, we excluded 156 records. Following this, we applied the inclusion criteria, which led to the exclusion of an additional 147 records. Finally, 81 records were included for analysis.

Figure 2 illustrates the distribution of publications per year, covering the period from 2019 to 2024. Among these, 25 studies were published in 2023, and 20 studies were published in 2022. In 2024, there were 13 studies, but, as this covers only the first 3 months until March 2024, we can assume that the number of review articles on the use of AI in IoT will continue to increase. Regarding earlier years, only 3 studies were published in 2019, 9 in 2020 and 13 in 2021.

### 2.3. Research Questions

The study aims to explore the general research question regarding the roles of Artificial Intelligence (AI) in Internet of Things (IoT) systems. In particular, it will address the following two research questions:RQ1: What are the different tasks that AI carries out in IoT systems?RQ2: Which are the application domains where AI has been used in IoT systems?

The first research question (RQ1) seeks to elucidate the various roles that AI plays in IoT systems. This includes understanding how AI enhances the functionality, efficiency and capabilities of IoT devices and networks through, e.g., advanced data analytics and autonomous decision-making. In order to identify the different categories of tasks, we first extracted the text describing the concrete task of the AI from the article. Then, we compared and elicited re-occurring tasks, grouping them under common more abstract labels, e.g., prediction, decision-making, etc.

The second research question (RQ2) focuses on identifying and categorizing the application domains where AIoT systems are utilized. These application domains may range from smart homes and healthcare to industrial automation, smart cities and beyond, highlighting the pervasive influence of AI in transforming IoT ecosystems across different sectors. By addressing these questions, the study aims to provide a comprehensive overview of the synergies between AI and IoT, revealing both the technological impacts and practical applications of their integration.

## 3. Results

During the review of the survey articles, we identified six different main categories of tasks for which AI methods have been used in IoT systems. By performing further analysis, we were able to identify a number of sub-categories for these main task categories. The following main and sub-categories were identified:Pattern recognition: This usually concerns the classification or identification of the current state, and subtasks include the following:–Event recognition (e.g., the recognition and detection of activities, intrusions, faults, anomalies);–Object recognition (e.g., the recognition and detection of humans, cars);–Diagnosis (e.g., medical, faults, malware detection);–Estimation (e.g., position of assets);–Authentication (e.g., devices, humans, products/food).Decision support: This is usually to help human users decide what action to take. Subtasks include the following:–Operational decisions (e.g., irrigation in farming);–Strategic decisions (e.g., city planning).Decision-making and acting: This concerns autonomous decision and actions by the IoT system. Subtasks include the following:–Resource allocation (e.g., task scheduling, task offloading, load balancing);–Control (e.g., lighting, temperature, vehicles, feeding fishes);–Planning (e.g., path planning of vehicles).Data management: This includes different types of data preprocessing. Subtasks include the following:–Reducing noise;–Removing irrelevant or sensitive data;–Filling in missing data.Prediction: Concerns the use of historical data to make predictions about future events, e.g., earthquakes.Human Interaction: This includes subtasks such as the following:–Natural language understanding (NLU);–Speech synthesis.

Figure 3 illustrates the various roles of Artificial Intelligence in Internet of Things (IoT) systems. The chart categorizes the AI roles and indicates the number of studies that focus on each role. Through the analysis, we found that pattern recognition is one of the main roles of AI in IoT systems and decision support is the second most popular task for which AI has been used. Table 1 provides more details of the 81 analyzed research articles about the role of AI in IoT systems.

Regarding RQ2, concerning the application domains where AI has been used in IoT systems, our analysis revealed that healthcare is the top AI application domain. With respect to application domains, as shown in the Figure 4, the following were identified in the study:Buildings: Includes the monitoring and management of different types of buildings like smart homes, office buildings, libraries and retail buildings.Emergency response: Includes fire evacuation, natural disasters, prediction and real-time response for improving efficiency and safety.Education: Includes intelligent education systems to support teachers and students, enhances personalized learning and streamlines administrative tasks through automation.Energy: Includes management of smart grids, for sustainable energy distribution.Environment: Includes air and water quality monitoring and control of environmental health and sustainability.Farming: Includes agriculture and aquaculture through precision farming, crop monitoring and automated farming equipment.Governance: Includes policy-making and public service.Healthcare: Includes healthcare at home, pandemic management and patient monitoring systems.Industry: Includes manufacturing processes, predictive maintenance and enhancing quality control.Logistics: Includes supply chain management, improved route planning and demand forecasting to enhance efficiency.Military: Includes decision-making support, surveillance and strengthening defense capabilities.Public safety: Includes real-time monitoring, emergency response and crime prediction through data analysis.Transportation: Includes autonomous vehicles, traffic management and route optimization.Waste management: Includes waste collection, recycling processes and monitoring of waste levels to enhance environmental sustainability.General: Some articles just mention potential application domains and do not refer to any particular domain.

The details of the 81 research articles pertaining to AI application domains as well as the specific focus of each study are presented in Table 2. It should be noted that, in order for a survey article to be categorized as covering a particular application domain, we required that the survey refer to at least one paper that has used AI in IoT for that domain. That is, it was not enough just to mention the application domain. Additionally, the analysis of the specific focus of each study is presented in Figure 5, which indicates that IoT security is a primary focus in the survey articles, followed by healthcare and the COVID-19 pandemic. Eighteen studies primarily focused on IoT security, making it the most prominent area of research. Healthcare is the second major focus, with 10 studies. Eight studies concentrated on IoT applications related to the COVID-19 pandemic. Similarly, three studies addressed IoT implementations in smart cities. In each of the other domains, Unmanned Aerial Vehicles (UAV), Explainable AI, food safety, the environment, network management and energy management, there were two studies.

The analysis of AI task frequency across various application domains is presented in Figure 6. For instance, in emergency response applications, three articles mentioned decision-making and acting, two articles mentioned decision support and two articles mentioned pattern recognition. Please note that an article can mention multiple tasks and application domains. The distribution of AI tasks such as decision-making and acting, decision support, pattern reorganization, data management and human interaction across different application domains reveals insightful trends. Certain AI tasks are more prevalent in specific application domains. For instance, the AI tasks of decision-making and acting are extensively applied across many application domains like healthcare, farming, transportation, industry and energy. Similarly, decision support is widely used in healthcare, farming, energy, the environment, industry, transportation and buildings.

Pattern recognition is mainly used in healthcare, industry, transportation, energy and the environment. Data management is used for healthcare, energy and farming most. This chart also highlights that some tasks like human interaction are less used across all the domains; it is only used for buildings. Overall, Figure 6 provides a clear view of which AI tasks are commonly associated with the different application domains and helps to identify the trends in AI applications.

### Common Tasks of AI for the Top Five Application Domains

The common tasks of AI for the identified top five application domains (i.e., healthcare, farming, energy, industry and transportation) are shown in Figure 7 and more details are provided in the following.

Healthcare: The reviewed articles indicate that AI is transforming the healthcare industry by enhancing both the efficiency and quality of the healthcare. AI IoT devices, such as wearables and smart sensors, improve diagnostics, automate routine tasks and make data-driven decisions. We found that pattern recognition is the primary AI task within healthcare, and that the most frequent subtasks mentioned in the articles are event recognition (50%), authentication (32%) and diagnosis (14%). Decision support is the second most common AI task in healthcare, where more articles focused on operational decisions (76%) than on strategic decisions (24%). The third most common task concerned decision-making and acting, mainly for resource allocation (64%) and control (36%). Finally, AI is used in healthcare also for data management, in particular for reducing noise (42%), removing sensitive data (33%) and filling in missing data (25%).Farming: Based on the analyzed articles, we identified farming as the second domain where AI is driving innovation and efficiency. Decision-making and acting emerged as the primary AI tasks, with focus on resource allocation (64%), control (27%) and planning (9%). Decision support, the second most common task of AI in farming, showed a strong frequency of operational decisions (91%) and only a few mentions of strategic decisions (9%). Data management was identified as the third most common task of AI in farming, including removing sensitive data (38%), filling in missing data (37%) and reducing noise (25%). We identified pattern recognition as the fourth most common AI task, including event recognition (50%), authentication (33%) and object recognition (17%).Energy: The third most prominent application domain that emerged from the analyzed articles was energy, where decision support was the main AI task and the only found subtask was operational decisions (100%). Pattern recognition is the second most common AI task, including event recognition (71%) and authentication (29%). Decision-making and acting was found to the third most common AI task, including resource allocation (83%) and control (17%). The fourth most common AI task, data management, includes the following subtasks: reducing noise (40%), removing sensitive data (40%) and filling in missing data (20%).Industry: Pattern recognition is the main AI task for industry, which is the fourth main application domain. The subtasks of pattern recognition are event recognition (27%), diagnosis (27%), authentication (27%), object recognition (13%) and estimation (6%). Decision support is the second most common AI task, with subtasks including operational decisions (86%) and strategic decisions (14%). The third most common AI task is data management, and its most common subtasks are filling in missing data (43%), reducing noise (29%) and removing sensitive data (28%). We identified decision-making and acting as the fourth most common AI task, with resource allocation (83%) and control (17%) as subtasks.Transportation: We identified transportation as the fifth most common application domain for AI, where pattern recognition is the primary AI task and subtasks are event recognition (40%), authentication (40%) and diagnosis (20%). Decision-making and acting is the second most common AI task here, with resource allocation (75%) and control (25%) being the subtasks. Decision support is the third most common AI task, with operational decisions (100%) being the only subtask mentioned by the analyzed articles. We identified data management as the fourth most common AI task with 33% each in reducing noise, removing sensitive data and filling in missing data.

## 4. Discussion

This systematic study reveals significant insights into and reflections on the role of AI in IoT systems. It offers a comprehensive understanding of the current state and potential future directions in this field of research. The integration of AI in IoT systems makes significant improvements to technologies. The roles of AI in IoT systems, like pattern reorganization, decision support, autonomous decision-making, data management, prediction and human interaction, enable IoT systems to function more intelligently and efficiently. AI autonomously performs complex tasks in IoT systems, such as resource allocation, control and planning. For instance, when we consider smart agriculture, AI within IoT systems can autonomously optimize resources, schedule irrigation based on real-time soil moisture data and also enhance the crop yield. This represents a significant milestone, a shift from human to fully autonomous systems, in the advancement of IoT applications. The other important insight relates to the prominence of AI in healthcare IoT systems, where pandemic management and patient monitoring have been significant achievements. The ability of AI to predict the medical conditions of patients, detect anomalies and support operational decisions in real time highlights its transformative impact on healthcare. In addition to healthcare, AI is also being applied in fields like smart building, education, emergency response, energy management, environmental monitoring, agriculture and governance, all aimed at enhancing operational efficiency.

Furthermore, our analysis also highlights the specific focus of 81 studies, and we identified that IoT security is a top concern, discussed in 18 studies. Security is critical given the increasing reliance on IoT systems across industries, and AI’s role in identifying vulnerabilities and responding to threats is a major area of research. While healthcare is the second most frequent focus of studies, we also found AI tasks frequently across 13 application domains, where decision-making and acting, decision support and pattern recognition having higher frequency across the healthcare and farming application domains. In conclusion, the integration of AI within IoT systems is reshaping industries by enabling smarter, more efficient and secure operations. The potential for further advancements in AIoT applications suggests promising future directions. These findings represent a valuable resource for researchers and practitioners in this field who are dedicated to innovating and implementing AIoT solutions, thereby fostering technological advancements and delivering societal benefits.

## 5. Conclusions

In this systematic mapping study, we analyzed 81 survey articles from multiple databases to identify the significant roles of AI in IoT systems. The results highlight the crucial roles of AI in IoT systems across various application domains, offering valuable insights for both researchers and practitioners in this field. We conclude that pattern recognition and decision support are the main tasks of AI in IoT systems, but also that autonomous decision-making and acting as well as data management and prediction are common tasks. Furthermore, healthcare emerged as the predominant application domain where AI is extensively employed in IoT systems. Other significant application domains are industry, farming, energy, transportation, buildings and the environment. The findings of this systematic mapping study provide a comprehensive understanding of the AI tasks and application domains in IoT systems.

## Figures and Tables

**Figure 1 sensors-24-06511-f001:**
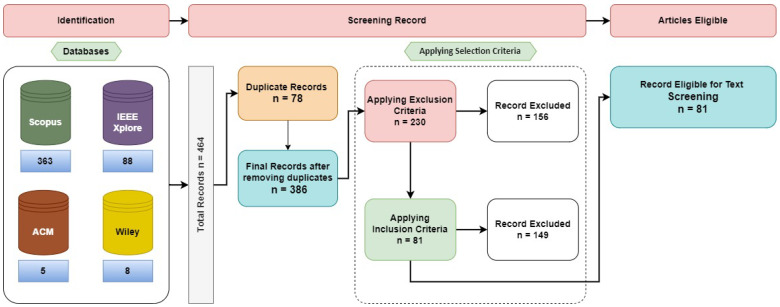
PRISMA flow diagram.

**Figure 2 sensors-24-06511-f002:**
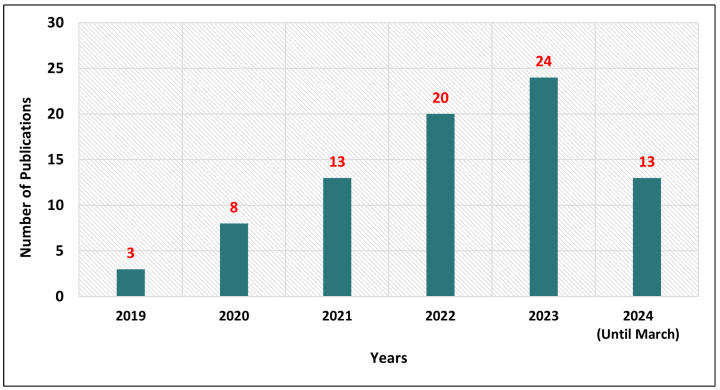
Publications per year.

**Figure 3 sensors-24-06511-f003:**
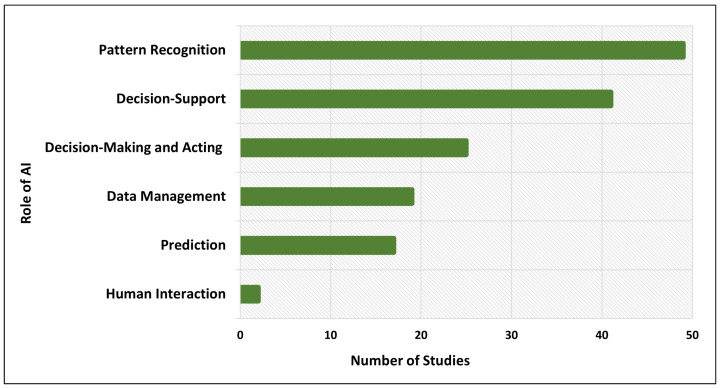
Role of AI in IoT systems.

**Figure 4 sensors-24-06511-f004:**
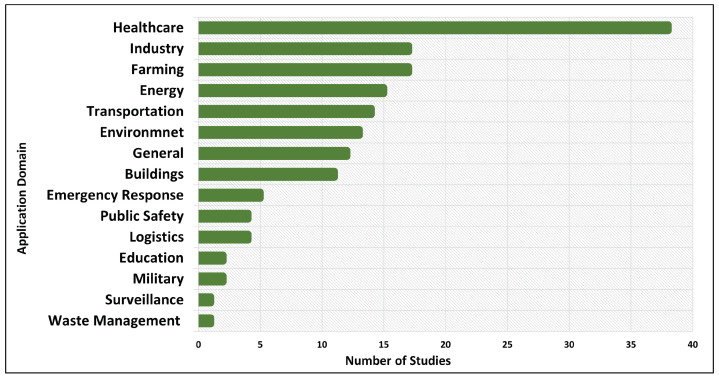
Applications domains of AI in IoT systems.

**Figure 5 sensors-24-06511-f005:**
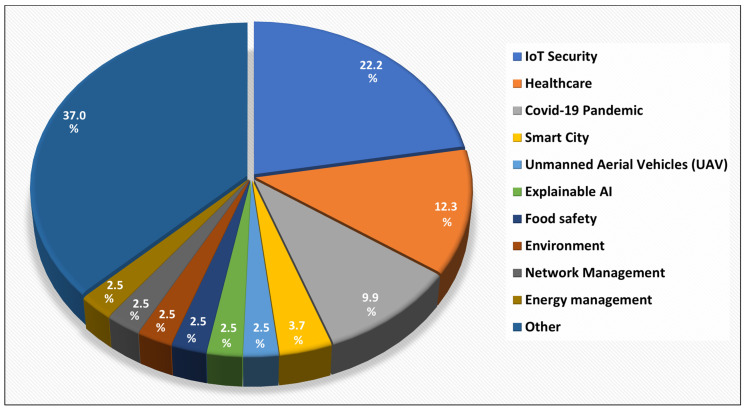
Specific focus of the studies.

**Figure 6 sensors-24-06511-f006:**
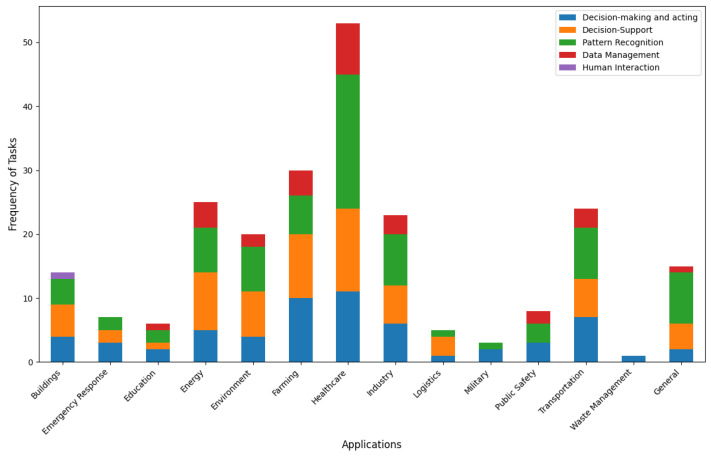
AI task frequency across different applications.

**Figure 7 sensors-24-06511-f007:**
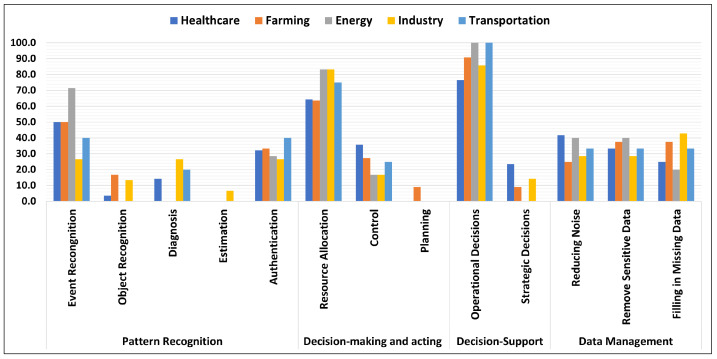
AI subtasks of the top five application domains.

**Table 1 sensors-24-06511-t001:** Role of AI in IoT systems.

Ref.	Decision-Making and Acting			Decision Support			Pattern Recognition					Data Management			Human Interaction	
	Resource Allocation	Control	Planning	Operational Decisions	Strategic Decisions	Prediction	Event Recognition	Object Recognition	Diagnosis	Estimation	Authentication	Reducing Noise	Removing Sensitive Data	Filling in Missing Data	NLU	Speech Synthesis
[5]	-	✓	✓	-	-	-	-	-	-	-	-	-	-	-	-	-
[6]	-	✓	-	✓	-	✓	✓	-	-	-	-	-	-	-	-	-
[7]	✓	-	-	-	-	✓	✓	-	-	-	-	✓	✓	✓	-	-
[8]	✓	✓	-	-	-	-	-	-	-	-	✓	✓	-	-	-	-
[9]	✓	✓	-	-	-	-	-	-	-	-	✓	✓	-	-	-	-
[10]	✓	-	-	-	-	-	✓	-	-	-	-	-	-	-	-	-
[11]	-	✓	-	-	✓	✓	-	-	-	-	-	-	-	-	-	-
[12]	-	-	-	-	-	✓	-	-	-	-	-	-	-	-	-	-
[13]	-	-	-	-	-	-	✓	-	-	-	✓	-	-	-	-	-
[14]	✓	-	-	✓	-	✓	-	-	-	-	-	-	-	-	-	-
[15]	✓	-	-	-	-	-	✓	-	-	-	-	✓	-	-	-	-
[16]	-	-	-	✓	-	-	-	-	-	-	-	-	-	-	-	-
[17]	✓	✓	-	-	-	-	-	-	-	-	-	-	-	-	-	-
[18]	-	✓	-	-	✓	✓	-	-	-	-	-	-	-	-	-	-
[19]	✓	-	-	-	-	-	-	-	-	-	✓	-	-	-	-	-
[20]	-	-	-	✓	-	-	-	-	-	-	-	✓	-	✓	-	-
[21]	-	-	-	✓	-	-	-	-	-	-	-	-	✓	✓	-	-
[22]	✓	-	-	✓	-	-	-	-	-	-	-	-	-	-	-	-
[23]	✓	-	-	-	-	-	-	-	-	-	-	-	-	-	-	-
[24]	-	-	-	✓	-	-	-	-	-	-	-	-	-	-	-	-
[25]	-	-	-	-	-	-	✓	-	-	-	-	✓	-	✓	-	-
[26]	-	-	-	-	-	✓	✓	-	-	-	✓	-	✓	-	-	-
[27]	-	-	-	✓	-	-	-	-	-	-	✓	-	-	-	-	-
[28]	-	-	-	-	-	-	-	-	-	✓	-	-	-	-	-	-
[29]	-	-	-	-	-	-	-	-	✓	-	✓	-	-	-	-	-
[30]	-	-	-	-	-	✓	-	-	✓	-	✓	-	-	-	-	-
[31]	✓	-	-	-	-	✓	✓	-	-	-	-	-	-	-	-	-
[32]	-	-	-	-	-	-	✓	-	-	-	✓	-	-	-	-	-
[33]	✓	-	-	-	-	-	-	-	-	-	-	-	-	-	-	-
[34]	-	-	-	✓	-	✓	-	-	-	-	-	-	-	-	-	-
[35]	-	-	-	✓	-	-	-	-	-	-	✓	-	-	-	-	-
[36]	-	-	-	-	✓	-	-	-	-	-	-	-	-	-	-	-
[37]	-	-	-	-	-	-	-	✓	-	-	-	-	-	-	-	-
[38]	-	-	-	-	✓	-	-	-	-	-	-	-	-	-	-	-
[39]	-	-	-	-	-	-	-	✓	✓	-	-	-	-	-	-	-
[40]	-	-	-	-	-	-	✓	-	-	-	-	-	-	-	-	-
[41]	-	-	-	✓	-	✓	-	-	-	-	-	-	-	-	-	-
[42]	-	-	-	-	-	-	✓	-	-	-	-	-	-	-	-	-
[43]	-	-	-	-	-	-	✓	-	-	-	-	-	-	-	-	-
[44]	-	-	-	✓	-	-	-	-	-	-	-	-	-	-	-	-
[45]	-	-	-	✓	-	-	-	-	-	-	-	-	-	-	-	-
[46]	-	-	-	-	-	-	✓	✓	-	-	-	-	-	-	-	-
[47]	-	-	-	✓	✓	✓	-	-	-	-	-	-	-	-	-	-
[48]	-	-	-	-	-	-	✓	-	-	-	-	-	-	-	-	-
[49]	-	-	-	-	-	-	✓	-	-	-	-	-	-	-	-	-
[50]	✓	-	-	✓	-	-	-	-	-	-	-	-	-	-	-	-
[51]	-	-	-	✓	-	-	✓	-	-	-	-	-	-	-	-	-
[52]	-	-	-	-	-	-	✓	-	-	-	-	-	-	-	-	-
[53]	-	-	-	-	-	✓	-	-	-	-	-	-	✓	-	-	-
[54]	-	-	-	-	✓	-	-	-	-	-	-	-	-	-	-	-
[55]	-	-	-	✓	-	✓	-	-	-	-	-	-	✓	-	-	-
[56]	-	-	-	✓	-	✓	-	-	-	-	-	-	-	-	-	-
[57]	-	-	-	-	✓	-	✓	-	-	-	-	-	-	-	-	-
[58]	-	-	-	✓	-	-	-	-	-	-	-	-	-	-	-	-
[59]	-	-	-	-	✓	-	-	-	-	-	-	-	-	-	✓	✓
[60]	✓	-	-	-	-	-	-	-	-	-	-	-	-	-	-	-
[61]	✓	-	-	✓	-	-	-	-	-	-	-	-	-	-	-	-
[62]	-	-	-	-	-	-	-	-	✓	-	-	-	-	-	-	-
[63]	-	-	-	-	✓	-	-	-	-	-	-	-	-	-	-	-
[64]	✓	-	-	-	-	-	-	-	-	-	-	-	-	-	-	-
[65]	-	-	-	✓	-	-	-	-	-	-	-	-	-	-	-	-
[66]	-	-	-	-	-	-	✓	-	-	-	-	-	-	-	-	-
[67]	-	-	-	-	-	-	✓	-	-	-	-	-	✓	-	-	-
[68]	-	-	-	✓	-	-	-	-	-	-	-	-	-	-	-	-
[69]	-	-	-	-	-	-	✓	-	-	-	-	-	-	-	-	-
[70]	-	-	-	✓	-	-	-	-	-	-	-	-	-	-	-	-
[71]	-	-	-	✓	-	-	✓	-	-	-	-	-	-	-	-	-
[72]	✓	-	-	-	-	✓	✓	-	-	-	-	-	-	-	-	-
[73]	-	-	-	✓	-	-	✓	-	-	-	-	-	-	-	-	-
[74]	-	-	-	✓	-	-	-	-	-	-	-	-	-	-	-	-
[75]	-	-	-	✓	-	-	-	-	-	-	-	-	-	-	-	-
[76]	-	-	-	-	-	-	-	-	-	-	-	✓	-	-	-	-
[77]	-	-	-	✓	-	✓	-	-	-	-	-	-	-	-	-	-
[78]	-	-	-	✓	-	-	-	-	-	-	-	-	-	-	-	-
[79]	-	-	-	✓	-	-	-	-	-	-	-	-	-	-	-	-
[80]	-	-	-	✓	-	-	✓	-	-	-	-	-	-	-	-	-
[81]	-	-	-	-	-	-	✓	-	-	-	-	-	-	-	-	-
[82]	-	-	-	✓	-	-	✓	-	-	-	-	-	-	-	-	-
[83]	-	-	-	-	-	-	-	-	-	-	✓	✓	-	✓	-	-
[84]	-	-	-	-	-	-	-	-	✓	-	✓	-	-	-	-	-
[85]	-	-	-	-	-	-	✓	-	-	-	-	-	-	-	-	-

**Table 2 sensors-24-06511-t002:** AI application domains in IoT systems.

Reference	Buildings	Emergency Response	Education	Energy	Environment	Farming	Healthcare	Industry	Logistics	Military	Public Safety	Transportation	Waste Management	General	Specific Focus
[5]	-	✓	-	-	-	✓	-	-	-	✓	✓	-	-	-	Unmanned Aerial Vehicles (UAV)
[6]	✓	-	✓	-	✓	✓	✓	✓	-	-	-	✓	-	-	Smart Cities (SC)
[7]	-	-	✓	-	✓	✓	✓	✓	-	-	-	✓	-	-	Edge Computing
[8]	-	-	-	✓	-	-	✓	-	-	-	✓	✓	-	-	Public Service
[9]	-	-	-	-	-	-	✓	-	-	-	✓	-	-	-	COVID-19 Pandemic
[10]	-	-	-	✓	✓	✓	✓	✓	-	-	-	-	-	-	Explainable AI
[11]	-	-	-	-	-	-	✓	-	-	-	-	-	-	-	Fall Prevention
[12]	-	✓	-	-	-	-	-	-	-	-	-	-	-	-	Prediction of Earthquakes
[13]	-	-	-	-	-	-	-	-	-	-	-	-	-	✓	IoT Security
[14]	-	-	-	-	-	✓	-	-	-	-	-	-	-	-	Agriculture
[15]	-	-	-	✓	-	-	-	-	-	-	-	-	-	-	Smart Grid
[16]	-	-	-	-	-	-	✓	✓	-	-	-	-	-	-	Explainable AI
[17]	-	-	-	-	-	-	✓	-	-	-	-	-	-	-	COVID-19 Pandemic
[18]	-	-	-	-	-	✓	-	-	-	-	-	-	-	-	Aquaculture
[19]	-	-	-	-	-	✓	✓	✓	-	-	-	✓	-	-	IoT Networks
[20]	-	-	-	-	-	✓	-	-	-	-	-	-	-	-	Agriculture and Food Industry
[21]	-	-	-	✓	-	✓	✓	✓	-	-	-	✓	-	-	Smart Sensors
[22]	-	-	-	-	-	✓	-	✓	-	-	-	-	-	-	Maritime, Aerial Systems and Industry 4.0
[23]	-	-	-	-	-	-	-	-	-	-	-	-	-	✓	Fog/Edge Resource Management
[24]	-	-	-	✓	-	-	✓	-	-	-	-	-	-	-	Smart Cities
[25]	-	-	-	-	-	-	-	-	-	-	-	-	-	✓	IoT Security
[26]	-	-	-	-	-	-	✓	-	-	-	-	-	-	-	Personalized Healthcare Services
[27]	-	-	-	-	-	-	-	-	-	-	-	-	-	✓	IoT Security
[28]	-	-	-	-	-	-	-	✓	-	-	-	-	-	-	Indoor Positioning
[29]	-	-	-	-	✓	-	✓	✓	-	-	-	✓	-	-	IoT Security
[30]	-	-	-	-	-	-	✓	-	-	-	-	-	-	-	Healthcare
[31]	✓	-	-	✓	-	-	✓	-	-	✓	-	✓	-	-	Wireless networks
[32]	-	-	-	-	-	-	-	-	-	-	-	-	-	✓	IoT Security
[33]	✓	-	-	✓	-	-	-	-	✓	-	-	✓	✓	-	Resource Efficiency
[34]	-	-	-	-	-	-	✓	-	-	-	-	-	-	-	COVID-19 Pandemic
[35]	✓	-	-	✓	-	✓	✓	-	✓	-	-	✓	-	-	IoT Security
[36]	-	-	-	-	-	-	-	✓	-	-	-	-	-	-	Industrial Needs
[37]	-	-	-	-	-	✓	-	-	-	-	-	-	-	-	Crop Disease Detection
[38]	✓	-	-	-	-	-	-	-	-	-	-	-	-	-	Strategic Decisions for Smart Homes
[39]	✓	-	-	-	-	-	✓	✓	-	-	-	-	-	-	IoT Security
[40]	-	-	-	-	-	-	✓	-	-	-	-	-	-	-	COVID-19 Pandemic
[41]	✓	-	-	✓	-	-	-	-	✓	-	-	-	-	-	Sustainability
[42]	-	-	-	-	-	-	✓	-	-	-	-	-	-	-	Heart Failure Monitoring
[43]	-	-	-	-	-	-	✓	-	-	-	-	-	-	-	COVID-19 Pandemic
[44]	-	-	-	-	-	✓	-	-	-	-	-	-	-	-	Grain Quality
[45]	-	-	-	-	-	-	-	-	✓	-	-	-	-	-	Fresh Food Logistics
[46]	-	-	-	-	-	-	-	✓	-	-	-	-	-	-	IoT Network Traffic Analysis
[47]	-	-	-	-	✓	-	✓	-	-	-	-	-	-	-	Air Quality
[48]	-	-	-	-	-	-	-	-	-	-	-	-	-	✓	IoT Security
[49]	-	-	-	-	-	-	✓	-	-	-	-	-	-	-	Diagnosis and Treatment Of Colorectal Cancer
[50]	✓	✓	-	-	✓	✓	✓	✓	-	-	-	✓	-	-	UAV
[51]	-	-	-	-	✓	-	✓	-	-	-	-	-	-	-	Environment and Health
[52]	-	-	-	-	-	-	✓	-	-	-	-	-	-	-	COVID-19 Pandemic
[53]	-	-	-	-	-	✓	-	-	-	-	-	-	-	-	Food Safety
[54]	-	-	-	-	-	-	✓	-	-	-	-	-	-	-	Healthcare
[55]	-	-	-	✓	-	-	-	-	-	-	-	-	-	-	Cognitive Sensing
[56]	-	-	-	-	-	-	-	-	-	-	-	-	-	✓	IoT Security
[57]	-	-	-	-	✓	-	-	-	-	-	✓	-	-	-	Environmental Pollution Monitoring and Management
[58]	-	-	-	-	-	-	-	✓	-	-	-	-	-	-	Process Management in Cyber-Physical Production Systems
[59]	✓	-	-	-	-	-	-	-	-	-	-	-	-	-	Libraries
[60]	-	-	-	-	-	-	✓	-	-	-	-	-	-	-	Sensing and Decision-Making
[61]	-	-	-	-	-	✓	-	-	-	-	-	-	-	-	Harvesting
[62]	-	-	-	-	-	-	-	✓	-	-	-	✓	-	-	Industrial IoT Security
[63]	-	-	-	-	-	-	✓	-	-	-	-	-	-	-	Healthcare Services for SC
[64]	-	-	-	-	-	-	-	-	-	-	-	-	-	✓	Network Management for SC
[65]	-	-	-	-	-	-	✓	-	-	-	-	-	-	-	COVID-19 Pandemic
[66]	-	-	-	-	-	-	✓	-	-	-	-	-	-	-	IoT Security
[67]	-	-	-	-	-	-	✓	-	-	-	-	-	-	-	Remote Healthcare Monitoring
[68]	-	-	-	✓	✓	✓	-	-	-	-	-	✓	-	-	Sensor Networks
[69]	-	-	-	-	-	-	-	-	-	-	-	-	-	✓	IoT Security
[70]	✓	-	-	✓	-	-	-	-	-	-	-	✓	-	-	Energy Management
[71]	-	-	-	✓	-	-	-	-	-	-	-	-	-	-	Energy Management
[72]	-	✓	-	-	-	-	-	-	-	-	-	-	-	-	Forest Fires
[73]	-	-	-	-	-	-	-	-	-	-	-	-	-	✓	IoT Security
[74]	-	-	-	-	-	-	✓	-	-	-	-	-	-	-	COVID-19 Pandemic
[75]	-	-	-	-	-	-	-	-	-	-	-	-	-	✓	Distributed Neural Networks
[76]	-	-	-	-	-	-	✓	-	-	-	-	-	-	-	Wearable Devices
[77]	-	-	-	-	-	-	✓	-	-	-	-	-	-	-	Spine Injuries
[78]	-	-	-	-	✓	-	-	-	-	-	-	-	-	-	Water Management
[79]	-	-	-	✓	✓	-	-	-	-	-	-	-	-	-	Smart Cities (SC)
[80]	-	-	-	-	-	-	✓	-	-	-	-	-	-	-	Healthcare Recommender Systems
[81]	-	-	-	✓	✓	-	-	-	-	-	-	✓	-	-	IoT Security
[82]	-	✓	-	-	-	-	-	-	-	-	-	-	-	-	Emergency Management
[83]	-	-	-	-	✓	-	✓	✓	-	-	-	-	-	-	Networking
[84]	✓	-	-	-	-	-	✓	✓	-	-	-	-	-	-	IoT Security
[85]	-	-	-	-	-	-	-	-	-	-	-	-	-	✓	IoT Security

## Data Availability

The replication package for this study includes the raw and processed data, and detailed documentation can be accessed at https://github.com/Umair1441/Role-of-AI-data (accessed on: 18 September 2024).

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
