# Peer review of "Exploring the Role of Artificial Intelligence in Internet of Things Systems: A Systematic Mapping Study"

_sensors, 2024, doi:10.3390/s24206511_

Round 1
Reviewer 1 Report
Comments and Suggestions for Authors
This manuscript presents a systematic mapping study that explores the application of Artificial Intelligence (AI) in Internet of Things (IoT). A total of 81 review publications from 2019 to 2024 were selected using the PRISMA approach. The in-depth analysis of these records, clearly illustrated through diagrams, graphs, and tables, highlights the domains where AI in IoT has been utilized and the tasks it has accomplished. These findings are valuable for researchers and practitioners across various domains in adapting AI-driven IoT solutions. Overall, the manuscript is well-written and well-presented.
However, while the study discusses the domains and tasks of AI in IoT separately, I recommend taking it a step further. To enhance the utility of this study for researchers and practitioners in each domain, it would be beneficial to include an analysis of the tasks within each domain, emphasizing their frequency and trends. For instance, if the researchers/practitioners from the healthcare industry are considering adopting AI in IoT, their questions could be (1) what are the commons tasks of AI in IoT in healthcare, and (2) which use cases are likely to succeed in healthcare? A holistic view of use cases across all domains, as shown in Table 1, may not provide sufficient insight. A detailed breakdown of use cases specifically for healthcare could be highly advantageous. While this manuscript briefly touches on this in the first paragraph of Section 4 Discussion, a more comprehensive analysis and discussion for additional domains, preferably with visualizations, would be desirable.
Comments on the Quality of English LanguageEnglish language looks good overall.
Author Response
Comment 1: However, while the study discusses the domains and tasks of AI in IoT separately, I recommend taking it a step further. To enhance the utility of this study for researchers and practitioners in each domain, it would be beneficial to include an analysis of the tasks within each domain, emphasizing their frequency and trends. For instance, if the researchers/practitioners from the healthcare industry are considering adopting AI in IoT, their questions could be (1) what are the commons tasks of AI in IoT in healthcare, and (2) which use cases are likely to succeed in healthcare? A holistic view of use cases across all domains, as shown in Table 1, may not provide sufficient insight. A detailed breakdown of use cases specifically for healthcare could be highly advantageous. While this manuscript briefly touches on this in the first paragraph of Section 4 Discussion, a more comprehensive analysis and discussion for additional domains, preferably with visualizations, would be desirable.
Response 1: Thanks for your comments and suggestions! The purpose of a systematic mapping study, which is a well-established scientific research methodology, is to categorize and classify existing research in a particular field. It should provide a broad overview of the research landscape, identifying areas that are well-studied and those that require further investigation [1]. We argue that this is what we do in the manuscript, and we have now added this explanation in the manuscript.
The deeper analysis that the reviewer asks for, is one of the goals of the systematic literature review research methodology, which would be an interesting future work. It is important to note that the requirement regarding the depth of analysis in a systematic mapping study is much lower compared to a systematic literature review.
However, to address your concern, we have added a section (3.1) that contains a more detailed analysis of the top five application domains.
Reviewer 2 Report
Comments and Suggestions for Authors
The paper analyzes the role of AI in Iot using database retrieval methods. Specific comments are as follows:
1. There is an error in line 97 “Finally, 81 records were included for analysis.” But the total number of publications in Figure 2 is 83.
2. The depth of analysis in this paper is not enough and does not give valuable insights
3. Line 122. The authors identified six different main categories of tasks for which AI methods have been used in IoT systems. But the authors do not state how these categories are determined. It is suggested that natural language processing methods be used to analyze the relevant articles.
4. There are significant problems with the citation of references in this article.
5. The links of Git provided by the authors are not accessible.
Comments on the Quality of English LanguageNo
Author Response
Comment 1: There is an error in line 97 “Finally, 81 records were included for analysis.” But the total number of publications in Figure 2 is 83.
Response 1: Thank you for pointing this out, now we corrected Figure 2 in the manuscript.
Comment 2: The depth of analysis in this paper is not enough and does not give valuable insights.
Response 2: The purpose of a systematic mapping study, which is a well-established scientific research methodology, is to categorize and classify existing research in a particular field. It should provide a broad overview of the research landscape, identifying areas that are well-studied and those that require further investigation [1]. We argue that this is what we do in the manuscript, and we have now added this explanation in the manuscript.
The deeper analysis that the reviewer asks for, is one of the goals of the systematic literature review research methodology, which would be an interesting future work. It is important to note that the requirement regarding the depth of analysis in a systematic mapping study is much lower compared to a systematic literature review.
However, to address your concern, we have added a section (3.1) that contains a more detailed analysis of the top five application domains.
Comment 3: Line 122. The authors identified six different main categories of tasks for which AI methods have been used in IoT systems. But the authors do not state how these categories are determined. It is suggested that natural language processing methods be used to analyze the relevant articles.
Response 3: Thanks, when analyzing the articles, we first extracted the text describing the task of the AI from the article, and then we compared and elicited re-occurring task grouping them under common more abstract labels, e.g. prediction, decision-making, etc. We have now explained this process in the manuscript, see line 122-125 in the manuscript.
Comment 4. There are significant problems with the citation of references in this article.
Response 4: Thank you for pointing this out, we have added the missing information of the references.
Comment 5. The links of Git provided by the authors are not accessible.
Response 5: Thank you for pointing this out, the Git hub link worked when accessing it on September 27, however now we update the link. https://github.com/Umair1441/Role-of-AI-data
Round 2
Reviewer 1 Report
Comments and Suggestions for Authors
Thanks for the response and revision.
Reviewer 2 Report
Comments and Suggestions for Authors
The authors addressed the issues raised by the reviewers and the quality of the article improved.